# A protocol to establish and maintain organotypic cerebellar slice culture (OCerSC) from aged mice

**Michael F. Almeida**[1]☯, **Kaitlan Smith** (ID)[1,2]☯, **Michael A. Garris**[1], **Rebekah Sanchez-Hodge** (ID)[1], **Meagan Colie** (ID)[1,2], **Jonathan C. Schisler** (ID)[1,2,3]*

**1** The McAllister Heart Institute, The University of North Carolina at Chapel Hill, Chapel Hill, North Carolina, United States of America, **2** Department of Pharmacology, The University of North Carolina at Chapel Hill, Chapel Hill, North Carolina, United States of America, **3** Department of Pathology and Lab Medicine, and Computational Medicine Program, The University of North Carolina at Chapel Hill, Chapel Hill, North Carolina, United States of America

☯ These authors contributed equally to this work
* schisler@unc.edu

## Abstract

Organotypic slice culture is a sophisticated technique historically used in cellular and developmental neurobiology to investigate three-dimensional architecture and cellular interactions. Maintaining thin tissue slices preserves organotypic structure and more accurately reflects the in vivo microenvironment than traditional monolayer cell cultures. These models are crucial in neurobiology, especially for studying age-dependent neurodegenerative disorders. However, preserving tissue viability in organotypic slices from adult brain tissue remains challenging. This protocol focuses on culturing organotypic slices from adult cerebellar tissue to address the critical need for discovery within this brain region. The cerebellum's unique cellular architecture and specialized circuitry have made culturing organotypic slices challenging. This protocol presents a novel finding: careful manipulation of culture conditions reduces early neuroinflammatory responses, enhances cerebellar tissue viability, and promotes the long-term maintenance of well-preserved cerebellar slices. This innovative system enables detailed analyses of neuronal morphology and functional connectivity at both cellular and circuit levels, incorporating advanced imaging and electrophysiological techniques to fulfill a critical need in neurobiology.

## Introduction

### Cell culture models to study neurodegeneration

Neurodegenerative diseases are age-dependent pathological disorders characterized by increased cell death, leading to progressive neuronal cell loss [1]. Patients commonly display a wide range of clinical symptoms, including cognitive impairment and motor coordination deficits [2,3]. Since the early 2000s, cell culture has become a standard method in neurobiology for modeling neurodegenerative diseases. This

Data availability statement: All relevant data are within the manuscript and its Supporting Information files.

Funding: 1. **Initials of the authors who received each award:** - R01AG066710 and R01AG061188: JCS - P30AG072958: MFA 2. **Grant numbers awarded to each author:** - JCS: R01AG066710, R01AG061188 - MFA: P30AG072958 3. **Full name of each funder:** - National Institute on Aging 4. **URL of each funder website:** - https://www.nia.nih.gov/ 5. **Role of sponsors or funders in the study:** - The sponsors or funders did not play any role in the study design, data collection and analysis, decision to publish, or preparation of the manuscript.

Competing interests: The authors have declared that no competing interests exist.

Abbreviations: Anova: Analysis Of Variation; AUC: Area under the curve; BSA: Bovine Serum Albumin; Cb: Cerebellum; COX: cyclooxygenase enzyme; Ct: cortex; DEV: days ex vivo; Hc: hippocampus; HS: horse serum; HS+I: horse serum plus indomethacin; LDH: lactate dehydrogenase; LSD: Least Significant Difference; LTD: long-term depression; MTT: 3-(4,5-dimethylthiazol-2-yl)-2,5-diphenyl-2H-tetrazolium bromide; NAD: nicotinamide adenine dinucleotide; NADH: Reduced Nicotinamide Adenine Dinucleotide; OCerSC: Organotypic Cerebellum Slice Culture; PBS: phosphate-buffered saline; PFA: paraformaldehyde; PVDF: polyvinylidene difluoride; ROI: region of interest; SF: serum-free; SF+I: serum-free plus indomethacin.

approach offers a stable, controlled environment with minimal systemic variation, advancing cellular and molecular neurobiology by enabling the studies of neuronal development, differentiation, survival, and physiology under genetic or pharmacological manipulation [4,5]. Despite these advantages, cell culture models, ranging from primary neurons to neuronal-like transformed cell lines, have significant limitations. These include the absence of an extracellular matrix, the requirement for anchorage-dependent monolayers, and insufficient gas exchange [6,7]. These inadequacies restrict their applicability and translational relevance, as evidenced by the limited efficacy of existing neurodegenerative therapies in clinical trials [8]. A comparison of the significant differences between cell culture and tissue culture is summarized in **Table 1**.

### The limitations of current models highlight the need for advanced solutions

Preclinical models are currently the most feasible systems for studying age-related neurodegenerative disorders. However, their high costs and limited control over chemical and physical complications present significant drawbacks. Therefore, models that more accurately represent the complexity of in vivo conditions are essential. These models can bridge the gap between preclinical animal models and humans, enabling a better understanding of pathological conditions and potentially leading to more translational outcomes.

### Organotypic brain slice culture

Organotypic brain slice culture has emerged as a powerful tool in neurobiology, serving as an alternative to monolayer cell cultures or preclinical models. This method preserves the brain's complex three-dimensional architecture and cellular composition ex vivo, providing a unique opportunity to investigate intricate neural processes. Various pathophysiological states can be simulated by directly manipulating culture conditions, enabling the study of neurodevelopmental disorders, neurodegenerative diseases, and mechanisms of synaptic plasticity.

### Challenges with culturing adult brain tissue

Despite its advantages, organotypic brain slice culture faces challenges, particularly in preparing slices and maintaining their viability [8]. Most studies use rodent brain slices from the first two neonatal weeks, as neonatal tissue demonstrates resistance to ischemic damage, immature metabolic states, and robust dendritic branching [9,10]. However, these characteristics do not accurately mimic age-related neuronal changes. Neonatal brain slices are characterized by resistance to ischemic damage [11], immature metabolic state [12], astrocytic differences [13], amplified synaptic plasticity [14], and robust dendritic branching [15]. To date, studies involving adult brain tissue have shown that slice viability is significantly lower than in neonatal tissue, a trend supported by ordinal logistic regression, which indicates that the odds of longer survival decrease with age. Specifically, juvenile slices (P22–P42) have approximately 16% odds of surviving longer than neonatal slices (P0–P14, OR =

**Table 1. Comparative Analysis of Cell Culture and Tissue Culture Techniques.** Summary of the advantages and disadvantages of cell culture and tissue culture techniques. Cell Culture refers to monolayer cell cultures, including primary, secondary, or cell lines, whereas Tissue Culture refers to three-dimensional tissue culture. A check mark (✓) indicates an advantage and a cross (×) indicates a disadvantage. Additionally, an asterisk (*) denotes considerations specific to primary cell lines, and a hashtag (#) signifies considerations specific to immortal cell lines.

| Considerations | Cell Culture | Tissue Culture |
|---|---|---|
| Consistent | ✓ | X |
| Reproducible | ✓ | X |
| Expenditure and Expertise | ✓ | X |
| Biological Material Generated | X | ✓ |
| Lifespan | X* | ✓ |
| Translational Outcomes | X | ✓ |
| Cell Heterogeneity | X | ✓ |
| Genetic Manipulation | ✓ | X |
| Chemical Manipulation | ✓ | X |
| Controlled Environmental | ✓ | ✓ |
| Growth Rate | ✓# | X |
| Cell-Cell Interactions | X | ✓ |
| Animal Use | ✓ | X |
| Risk of Contamination | ✓ | X |
| Chromosomal Instability | X | ✓ |
| Mimic in vivo Conditions | X | ✓ |

0.155, $p = 0.0778$). In comparison, adult slices (>P42) have approximately 9% odds (OR = 0.090, $p = 0.0176$), reflecting a significant reduction in viability with increasing age (**Fig 1**, **Table 2**). This corresponds to a median of 7 days ex vivo across all brain regions in adult tissue, compared with 22 days in neonatal tissue [9,16–24]. These findings highlight the need for long-term organotypic brain slice cultures from mature brains to support age-related studies, a gap that our protocol addresses, demonstrating 14-day viability (**Fig 1**).

## Focus on cerebellar tissue

Since their introduction in 1966 [25], organotypic cultures have been developed from various brain regions [8], but the cerebellum remains relatively underexplored. The cerebellum's unique laminar architecture and specialized circuitry pose significant challenges for culturing organotypic slices [26]. Its cortex comprises three distinct layers: the molecular layer, the purkinje cell layer, and the granule cell layer. Excitatory granule cells receive external inputs and project their axons to connect with the dendrites of inhibitory purkinje cells in the molecular layer. purkinje cells, in turn, relay signals to the cerebellar nuclei, which connect to the brainstem, spinal cord, and other brain regions. Adult cerebellar tissue exhibits reduced neuroplasticity and heightened sensitivity to environmental changes, requiring careful optimization of culture conditions to preserve viability, structural integrity, and functional connectivity [26].

## A protocol to enhance cerebellar slice viability from the adult brain

This protocol effectively maintains ex vivo culture of cerebellar brain slices from aged rodents (older than 6 months) for at least 14 days by modifying the medium composition (**fig 1**, **2a**). Key modifications include the use of serum-free media [18] and the anti-inflammatory compound indomethacin [27–31]. These changes improve cerebellar tissue viability and

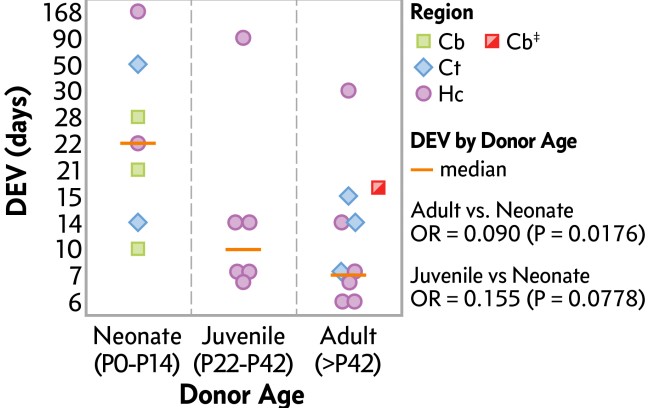

**Fig 1. The lifespan of organotypic brain slice cultures decreases with increasing donor age.** A metadata analysis of 21 published studies reveals that cultured rodent brain slices ex vivo, using 30 mm membrane inserts in 6-well plates, are represented as a dot plot, with days ex vivo (DEV) plotted against donor developmental age (days). Symbols indicate the brain region used for slice culture: (□) Cb = cerebellum, (◇) Ct = cortex, (○) Hc = hippocampus, (▨) Cb‡ = data from current study as reference and excluded from statistics. Horizontal lines represent median DEV values. To assess the relationship between DEV (ordinal outcome) and donor age group (neonatal, juvenile, adult), independent of brain region, ordinal logistic regression was performed. The model included donor age as the independent variable and DEV as the dependent variable. Model fit was assessed using a likelihood ratio test ($X^2 = 6.7$, $P = 0.0346$), and odds ratios were calculated to estimate the effect of donor age on culture longevity. The analysis confirmed that cultures from adult brain tissue exhibit significantly reduced viability, with a median DEV of 7 days across brain regions. In contrast, cultures from neonatal brain tissue have a median DEV of 22 days. The references for the included studies are listed in Table 2.

preserve arborization in cerebellar slices, as demonstrated by biochemical and imaging analyses, thus enabling comprehensive analysis of neuronal morphology and function.

## Materials and methods

### Euthanasia

Mice were euthanized if they exhibited critical health deterioration, including head tucking, skin lesions, malocclusions, abnormal breathing, urinary issues, self-mutilation, inability to walk, eat, or drink, agonal breathing, severe ulcerations, or uncontrolled bleeding. In cases of reduced mobility that limited access to food or water, hydrogel and moistened chow were placed on the cage floor. Mice were anesthetized and euthanized using isoflurane (1–3% in oxygen, drop method, as needed) via inhalation, followed by cervical dislocation to confirm death. Death was verified by testing footpad reflexes, in accordance with the recommendations of the American Veterinary Medical Association's Panel on Euthanasia.

### Cerebellar culture

The protocol described in this peer-reviewed article is published on protocols.io (dx.doi.org/10.17504/protocols.io.q26g-71428gwz/v1) and included as Supporting Information File 1 at the end of this document. This protocol assessed the effects of media composition on maintaining adult organotypic cerebellar slice cultures (OCerSC) ex vivo for up to 16 days (Fig 2A). Dissection and slicing were performed in ice-cold Hibernate-A medium (Gibco), supplemented with B27 and Glutamax, to minimize excitotoxicity and preserve tissue viability at ambient $CO_2$ levels without continuous carbogen gassing. Hibernate-A is specifically formulated for short-term storage and handling of postnatal/adult neural tissue, providing stable osmotic and physiological support during benchtop procedures and reducing mechanical stress on fragile, aged cerebellar slices. For the first five days, slices were cultured under four conditions: with or without serum, and with or without the anti-inflammatory agent indomethacin. Starting on day six, all slices were transferred to a common maintenance medium, with or without serum, and maintained through day 16.

**Table 2. Summary of Organotypic Slice Culture Experiments and Outcomes by Brain Region.** Data from various studies on organotypic slice cultures detailing the age of the rodents (weeks, w), days ex vivo (DEV), rodent model used, and references with DOIs. The studies are categorized by brain regions: Hippocampus, Cortex, and Cerebellum (Cbl).

| Region | Age (w) | DEV | Rodent Model | Reference | DOI |
|---|---|---|---|---|---|
| **Hippocampus** | 22.5 | 14 | Mouse | [9] | 10.3390/cells12101422 |
| | 3 | 7 | Rat | [24] | 10.1016/j.brainres.2003.12.009 |
| | 4.5 | 14 | Rat | [46] | 10.1016/s0165-0270(03)00228-0 |
| | 7 | 6 | Rat | [23] | 10.1177/026119290203000304 |
| | 5 | 7 | Mouse | [21] | 10.1016/j.brainres.2011.01.115 |
| | 4.5 | 14 | Rat | [20] | 10.1016/j.brainresprot.2004.03.004 |
| | 10 | 6 | Rat | [19] | 10.1016/j.brainresbull.2010.10.008 |
| | 13.5 | 30 | Mouse | [18] | 10.1016/j.pnpbp.2012.11.004 |
| | 1.6 | 168 | Mouse | [47] | 10.1385/JMN:19:3:317 |
| | 47 | 7 | Mouse | [16] | 10.1016/j.mex.2017.03.003 |
| | 3 | 90 | Rat | [10] | 10.1016/s0165-0270(00)00197-7 |
| | 90 | 7 | Mouse | [48] | 10.15252/embj.201694591 |
| | 3 | 7 | Rat | [24] | 10.1016/j.brainres.2003.12.009 |
| | 1.6 | 22 | Rat | [49] | 10.1111/bpa.12936 |
| **Cortex** | 45 | 15 | Mouse | [50] | 10.1371/journal.pone.0045017 |
| | 1 | 50 | Mouse | [22] | 10.1371/journal.pone.0022040 |
| | 7 | 7 | Mouse | [22] | 10.1371/journal.pone.0022040 |
| | 1 | 14 | Rat | [51] | 10.1016/j.brainres.2004.11.014 |
| | 40.5 | 14 | Mouse | [17] | 10.3389/fnagi.2015.00047 |
| **Cbl** | 0.15 | 21 | Mouse | [52] | 10.1080/14734220600905317 |
| | 2 | 28 | Mouse | [53] | 10.1615/critrevneurobiol.v18.i1-2.180 |
| | 1 | 10 | Mouse | [54] | 10.1016/s1385-299x(03)00020-5 |

### Cell cytotoxicity assay

A lactate dehydrogenase (LDH) assay was used to evaluate cytotoxicity in organotypic cerebellar slice cultures (OCerSC). LDH is a stable cytosolic enzyme released when the cell membrane loses integrity, serving as a dependable marker of cytotoxicity [32]. We used the LDH-Glo™ Cytotoxicity Assay (Promega, J2380), a luminescent test that measures LDH activity through linked enzymatic reactions, ultimately producing a luciferin-based bioluminescent signal that correlates with LDH levels.

To ensure precise and specific measurement of tissue-derived LDH release, the following controls and normalizations were applied:

- **Background correction:** LDH activity was measured in cell-free media samples (identical composition, incubated under the same conditions) and subtracted from experimental values to account for any intrinsic LDH-like activity in the reagents or media components.

- **Serum interference control:** Baseline LDH levels were determined in fresh serum-free and horse serum-supplemented media formulations before use.

- **Normalization to maximal release:** To express cytotoxicity as a percentage of total releasable LDH, parallel control wells (triplicate OCerSC at ~14 DIV) were treated with 10 × lysis solution (provided in the kit) or 10% Triton X-100 for 30–60 min to induce complete cell lysis. Media from these wells were collected and processed identically.

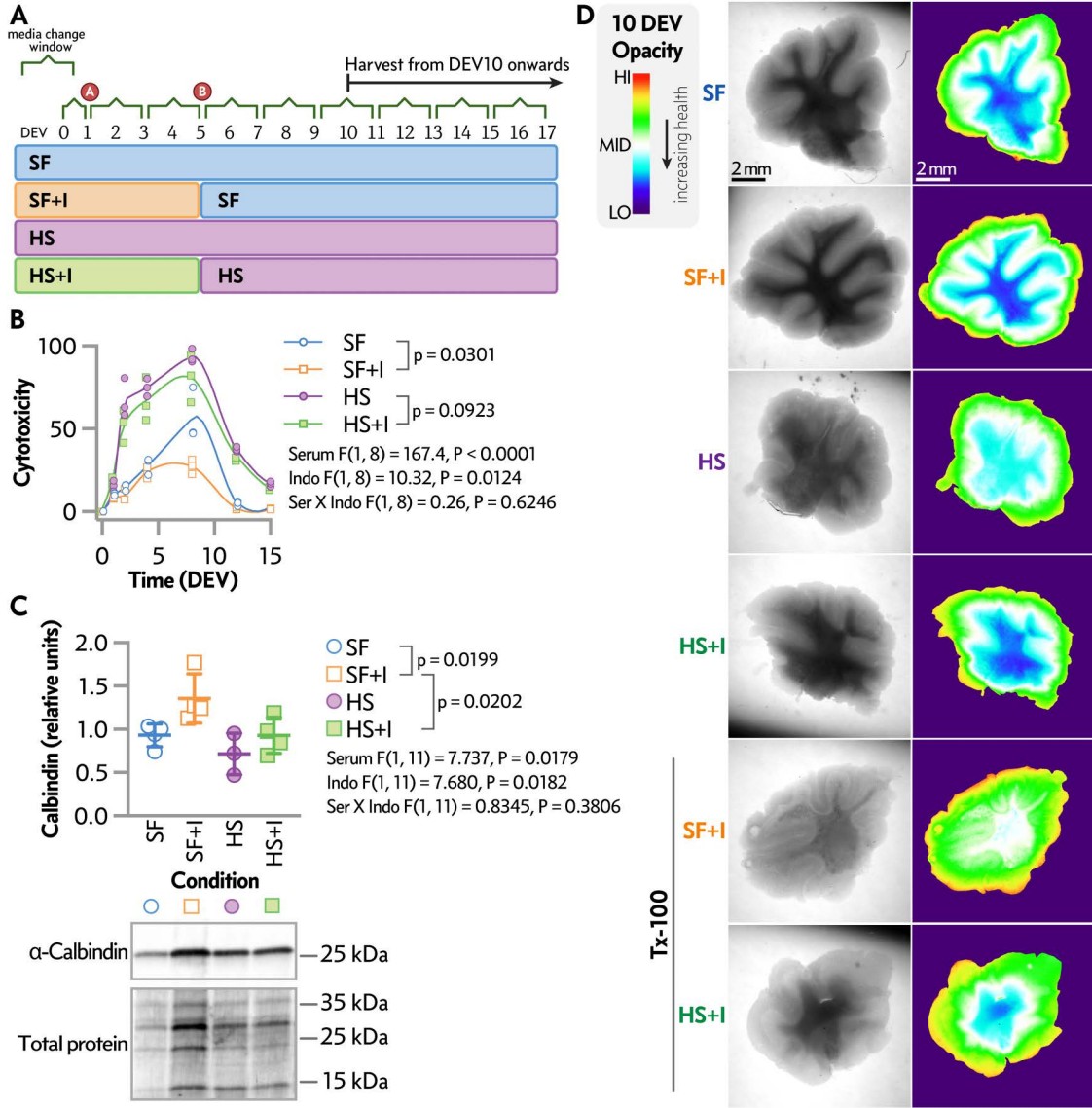

**Fig 2. Extended maintenance of OCerSC from aged rodent brain tissue.** (A) Timeline of the culture protocol used to maintain OCerSC from adult rodents for up to 16 days ex vivo (DEV). Slices were initially cultured in Slice Culture Media #1 (with or without serum or indomethacin) for five days under four conditions: horse serum (HS), horse serum plus indomethacin (HS + I), serum-free medium (SF), or serum-free medium with indomethacin (SF + I). On day six, all groups switched to Slice Culture Media #2 (free of indomethacin, with or without serum) and were maintained through day 16, with media changes every 48 hours. The circled points A and B indicate key timepoints, including the first media change at 18–24 hours and the transition to the maintenance medium on day six. Illustration created with BioRender.com. (B) Cytotoxicity, measured by LDH release over time in the SF + I and SF conditions (using the optimized protocol; see Methods), is shown as a spline line plot with individual data points overlaid and summarized as mean ± SD at each time point (N = 3 biological replicates). LDH release was lowest in the SF + I condition. Compared with HS, both HS + I and SF reduced LDH release; however, the greatest reduction was observed with SF + I, which decreased cytotoxicity by 71% relative to HS and 52% relative to SF (based on area under the curve calculations; see Supplementary Table 1). Statistical analysis of LDH time-course data and AUC values using two-way ANOVA (factors: serum and indomethacin) is shown in-panel and detailed in Supplementary Table 1. (C) Densitometry analysis (upper) and representative Western blot (lower) of calbindin expression at 14 DEV across the four initial conditions: SF, SF + I, HS, and HS + I. Densitometry values are shown as individual data points with mean ± SD (N = 4 biological replicates). Calbindin levels were highest in the SF + I condition. Relative calbindin protein levels were analyzed using two-way ANOVA (factors: serum and indomethacin); full statistical results are presented in the panel. Molecular weight markers in kilodaltons (kDa) are indicated for the calbindin Western blot and the total protein on the membrane (Stain-Free) used for normalization. (D) Bright-field images taken at 10 days ex vivo (DEV) to assess cerebellar tissue health under different conditions. A rainbow lookup table (LUT) was applied to the images, where color represents tissue opacity as follows: red = high opacity (less transparent, more degenerated tissue), white = intermediate, dark

blue = low opacity (highly transparent, healthier tissue). SF + I slices showed the greatest flattening and transparency, indicated by the largest dark blue regions representing well-preserved central cerebellar lobules. The HS + I condition showed moderate improvement over HS alone, but less than SF + I. These results were confirmed by treating a representative SF + I slice with Triton X-100, which caused delipidation and structural degradation, shifting the color toward higher opacity, serving as a positive control for cytotoxic damage. Overall, initial culture in SF + I provided the best long-term preservation of cerebellar structure.

- **Contamination prevention:** All culture media were supplemented with penicillin-streptomycin to minimize bacterial contributions to spurious LDH signals.

   Media sampling and processing: At each scheduled full media change (days 1, 2, 4, 8, 12, and 15), 5 µL of conditioned medium was collected from each culture insert and immediately diluted into 95 µL of ice-cold LDH storage buffer (200 mM Tris-HCl pH 7.3, 1% BSA, 10% glycerol) and stored at −20 °C until batch analysis. For the assay, technical triplicates were prepared by mixing 12.5 µL of diluted sample with 12.5 µL of LDH Detection Reagent in white 384-well plates. Reactions were incubated at room temperature for 45 min (protected from light), and luminescence was read on a CLARIOstar plate reader (BMG Labtech). Data are presented as background-corrected relative light units (RLU) or as a percentage of maximal LDH release where indicated.

### Western blot and chemifluorescence analysis

Calbindin protein expression in organotypic cerebellar slice cultures (OCerSC) was evaluated using Western blot analysis. Three slices per replicate (N = 4) were lysed in RIPA buffer supplemented with Halt Protease and Phosphatase Inhibitor Cocktail (Thermo Scientific, 78447). Protein lysates were separated on 4–15% Mini-PROTEAN TGX Stain-Free gels (Bio-Rad, 4568084) and transferred to PVDF membranes (Bio-Rad, 1620264). Membranes were blocked in 5% non-fat milk in TBST for 1 hour at room temperature, and incubated overnight at 4°C with calbindin (D1I4Q) XP® Rabbit mAb (1:1000, Cell Signaling Technology, #13176). Subsequently, membranes were incubated with StarBright Blue 700 secondary antibody (Bio-Rad, 12004161) for 1 hour at room temperature. Chemifluorescence signals were detected using a ChemiDoc Imaging System (Bio-Rad). Densitometry analysis was performed using Bio-Rad Image Lab software (version 6.1), with calbindin band intensity normalized to total protein content determined by stain-free membrane quantification.

### OCerSC light intensity imaging

To evaluate OCerSC health, bright-field images were captured using the Leica Stereo Microscope MZ 16 with the ORCA-ER CCD camera. Images were processed using Fiji [33]. A thermal lookup table (LUT) was applied during processing to highlight opacity variations across slices. To focus on a specific OCerSC area (designated as the region of interest, ROI), the area outside the ROI was excluded for clearer visualization.

### Immunofluorescence staining and imaging

Organotypic cerebellar slice cultures were fixed in 4% paraformaldehyde (PFA) for 20–24 hours at 4 ºC. Slices were then rinsed three times with phosphate-buffered saline (PBS) and either stored in PBS at 4 ºC or processed for immunofluorescence staining. Slices were permeabilized in PBS with 0.5% Triton-X100 for 30 minutes at room temperature, blocked with a 5% BSA solution in PBS with 0.1% Triton-X100 for 90 minutes at room temperature, and incubated with primary antibodies targeting calbindin (D1I4Q; Alexa fluor 488, Cell Signaling Technology #65152S) diluted in a 3% BSA solution containing 0.1% Triton-X100 for 48 hours at 4 ºC. After the primary antibody incubation, slices were rinsed three times with PBS containing 0.1% Triton-X100. During the second wash, NucBlue™ Fixed Cell ReadyProbes™ Reagent (Invitrogen, MA, USA) was added to label nuclei. Slices were mounted onto glass slides and covered with #1.5 coverslips using Prolong™ Diamond Antifade Mountant (Thermo Fisher Scientific, NC, USA), and imaged using an Andor Dragonfly spinning disk

confocal with the Zyla Plus 4.2MP sCMOS (Oxford Instruments, Abingdon, United Kingdom), Leica STELLARIS 8 FAL-CON, or Leica Mica in confocal mode. Images were processed using Imaris Multidimensional Visualization and Analysis software (Schlieren-Zurich, Switzerland) and Fiji, as specified.

## Statistical analysis

All statistical analyses were performed using JMP Pro (version 19, SAS Institute, Cary, NC, USA). To examine the relationship between days ex vivo (DEV, ordinal outcome) and donor age group (neonatal, juvenile, adult [34]), independent of brain region, ordinal logistic regression was used (N = 22, 6–9 per donor age group). The model included DEV as the dependent variable and donor age as the independent variable. Model fit was assessed using the likelihood ratio test, and odds ratios were calculated to quantify the effect of age on DEV.

For cytotoxicity and protein expression outcomes, the area under the curve (AUC) of LDH from biological replicates (N = 3 per condition) and relative calbindin expression from Western blots (N = 4 biological replicates, 3 slices per replicate) were analyzed using two-way analysis of variance (ANOVA). The 2x2 factorial design included two main effects-serum (horse serum [HS] vs. serum-free [SF]) and indomethacin (with vs. without)- and their interaction term. Calbindin expression was quantified by densitometry, with band intensity normalized to total protein content. Post hoc comparisons were conducted using Fisher's Least Significant Difference (LSD) test to identify specific group differences. Assumptions of normality and homoscedasticity were verified using Shapiro-Wilk and Levene's tests, respectively. All tests were two-tailed with a significance threshold of $\alpha = 0.05$. Data are reported as means ± standard deviation.

## Ethics declarations

All animal work was conducted in accordance with the Guide for the Care and Use of Laboratory Animals under approved IACUC animal use protocols within the AAALAC-accredited program at the University of North Carolina at Chapel Hill. NIH/PHS Animal Welfare Assurance Number: D16-00256 (A3410-01); USDA Animal Research Facility Registration Number: 55-R-0004; and AAALAC Institutional Number: #329. All animals were euthanized in compliance with the governing protocol.

## Expected results

By following this protocol (Supporting Information File 1), researchers can maintain organotypic cerebellar slice cultures (OCerSC) from aged cerebellar tissue (donors over 6 months old) ex vivo (DEV) for at least 14 days. The modified serum-free culture medium supplemented with indomethacin (SF + I) reduces cytotoxicity by 71% compared to standard horse serum (HS) conditions and 52% compared to serum-free (SF) conditions, as shown by lactate dehydrogenase (LDH) release assays (**Fig 2B, Supplementary Table 1 in S1 File**). This substantial reduction in cytotoxicity is critical for sustaining tissue viability over extended periods. Notably, cytotoxicity levels in OCerSC under SF + I conditions, with continuous indomethacin supplementation over the 16-day ex vivo period, remain comparable to those observed during the initial five days of culture (**Supplementary Fig 1**). These findings suggest that the protective effects of indomethacin occur primarily within the first five days, after which tissue viability stabilizes despite ongoing treatment.

The protocol preserves calbindin content at 14 days (DEV), as quantified by Western blot analysis (**Fig 2C**), indicating enhanced preservation of Purkinje cells. The cerebellum's intricate architecture, including the laminar organization (**Fig 2D**), densely packed granule cell layer, elaborate Purkinje cell dendritic arbors, and intraneuronal networks, is maintained, as demonstrated by calbindin staining at 14–16 DEV (**Fig 3**). At 7 DEV, OCerSC under SF + I conditions show minimal degeneration and retain dense granule cell layers. By 14 DEV, tissue integrity remains intact, with preserved Purkinje cell morphology and connectivity (**Fig 3**).

This protocol enables advanced biomolecular analyses of aged cerebellar tissue, including Western blotting for protein expression, immunofluorescence for cytoarchitectural evaluation, and enzyme-linked assays of collected

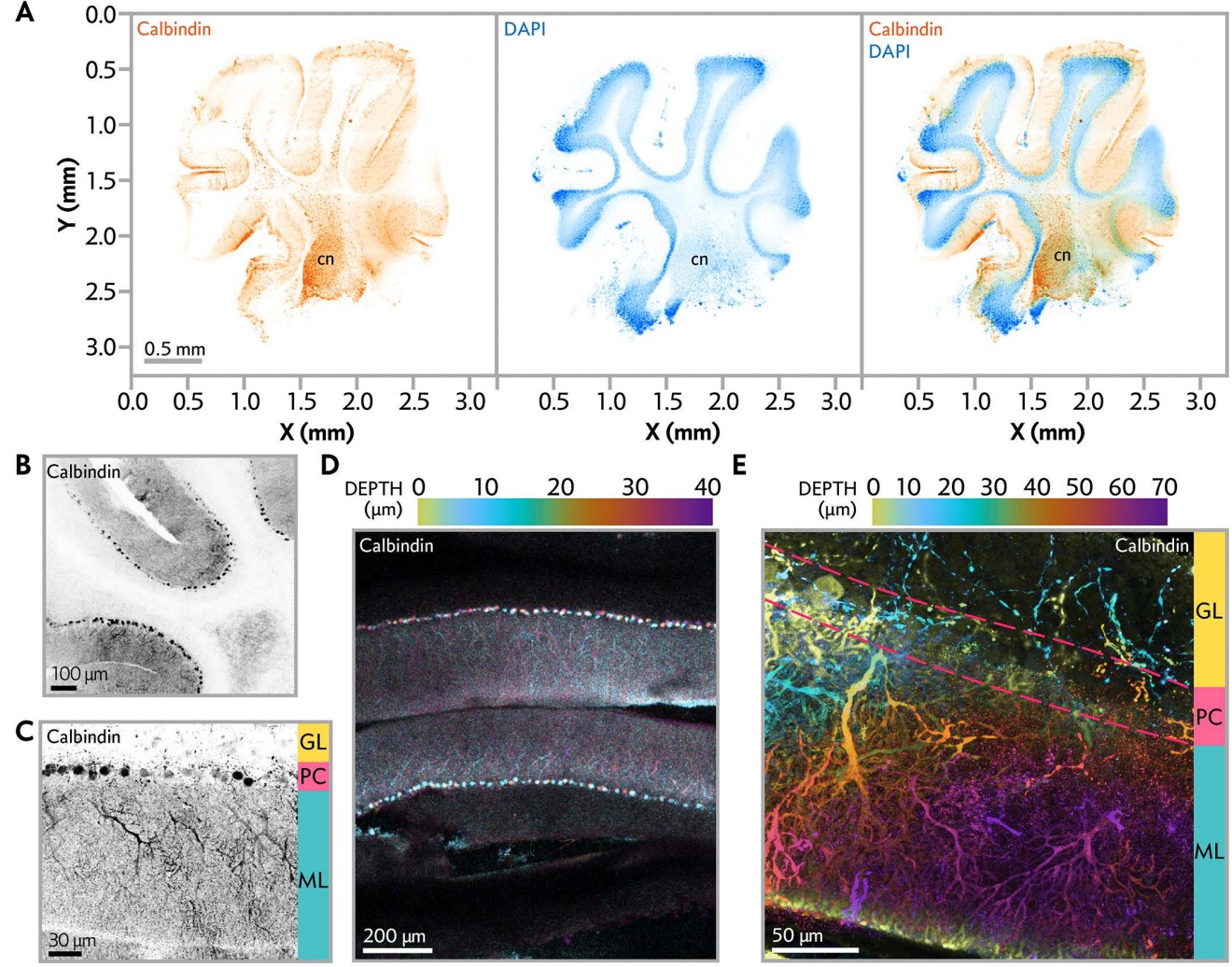

**Fig 3. Structural and neuronal integrity of OCerSC maintained in SF + I conditions.** OCerSC from aged rodents was cultured for the first 5 days ex vivo in serum-free medium supplemented with indomethacin (SF + I), followed by serum-free medium alone until fixation at 14–16 days ex vivo (DEV). Slices were fixed in 4% paraformaldehyde at 4°C for 24 hours before immunostaining and imaging. (A) Low-magnification overview of a representative SF + I-maintained tissue slice immunolabeled for calbindin (a Purkinje cell marker, green) and counterstained with NucBlue (nuclei, blue). The SF + I condition maintained the overall cerebellar cytoarchitecture, including distinct lobular organization and deep cerebellar nuclei (labeled "cn"). (B–E) High-magnification confocal images demonstrating preserved laminar organization and neuronal morphology in SF + I cultures. Calbindin immunostaining reveals healthy Purkinje cell somata aligned in a monolayer within the Purkinje cell layer, a densely packed granule cell layer beneath, and elaborate Purkinje cell dendritic arbors extending into the molecular layer. NucBlue counterstaining (blue) highlights intact nuclear organization throughout the cortical layers. These features confirm the maintenance of characteristic in vivo-like cerebellar architecture and synaptic connectivity during prolonged ex vivo culture. (D, E) Maximum-intensity z-stack projections with depth-encoded coloring of calbindin-positive structures. Green indicates superficial planes (0 μm), transitioning to purple in deeper planes (covering 0–40 μm in D and 0–70 μm in E). This visualization highlights the three-dimensional complexity and spatial preservation of Purkinje cell dendrites within the molecular layer.

culture media. When successfully implemented, it produces organotypic cerebellar slice cultures (OCerSC) that stay viable and structurally intact for at least 15 days in vitro (DIV), offering a valuable ex vivo model of mature cerebellar circuits.

Researchers can expect the following outcomes when following the optimized serum-free + initial indomethacin (SF + I) condition:

- **Cytotoxicity monitoring via LDH release:** LDH levels in culture media, measured using a luminescent assay (e.g., LDH-Glo™), are low and stable throughout the culture period under SF + I conditions (**Fig 2B**). Cytotoxicity typically peaks mildly during the early recovery phase (first 4–5 DIV) and declines thereafter, reflecting effective tissue healing and minimal ongoing cell death. Prolonged indomethacin exposure beyond the initial 5 DIV does not further reduce cumulative LDH release (**Supplementary Figure 1)**.

- **Preservation of cerebellar cytoarchitecture:** By 10–15 DIV, slices flatten well and exhibit clear lobular organization. Immunofluorescence staining reveals numerous healthy Purkinje cells aligned along lobules (strong calbindin signal) and a dense, intensely DAPI-stained granular layer (**Fig 3**), confirming excellent maintenance of neuronal populations and overall tissue integrity.

- **Improved optical properties:** Compared to acutely prepared slices, cultured OCerSC display reduced light scattering and enhanced transparency by ~10 DIV (**Fig 2D**). This progressive clearing facilitates high-resolution confocal or wide-field imaging deep into the tissue.

These outcomes demonstrate the protocol's dependability in maintaining the viability and function of cerebellar slices from aged mice. Slight variability in baseline LDH release may occur due to biological differences among older donors, but the positive effect of the SF + I condition remains consistent across replicates. When executed properly, researchers should achieve robust, reproducible OCerSC suitable for exploring age-related cerebellar biology, neuroprotection, and disease mechanisms.

## Discussion

This protocol addresses critical methodological gaps in neurobiology by enabling the long-term maintenance of adult three-dimensional cerebellar tissue cultures while preserving aging-related structural and functional changes. By extending the viability of OCerSC from aged donors (over six months old) to at least 14 DEV, our method overcomes the limitations of existing protocols, which typically fail to sustain brain explants beyond 7 days due to tissue degeneration (**Fig 1**) [27]. This advancement facilitates comprehensive studies of cellular dynamics, neuronal network activity, and drug responses in mature neuronal systems under controlled conditions.

The serum-free medium supplemented with indomethacin (SF + I) reduces cytotoxicity while preserving the cerebellum's native architecture, which is essential for studying neurobiology related to motor coordination and cognitive processing. These findings align with prior research demonstrating that serum-free conditions preserve neuronal viability and tissue architecture [18] and that indomethacin mitigates inflammation by inhibiting cyclooxygenase enzymes (COX-1 and COX-2), thereby reducing prostaglandin synthesis [35,36]. By controlling initial post-slicing inflammation, our protocol supports tissue healing while preventing prolonged neuroinflammatory signaling that can compromise viability.

This protocol provides a robust platform for investigating cerebellar contributions to aging-related neurological processes, such as motor coordination deficits and cognitive impairments observed in aging populations. The higher calbindin expression in SF + I conditions (**Fig 2C**) underscores the protocol's ability to maintain Purkinje cell health, supporting studies of Purkinje cell dysfunction implicated in age-related cerebellar ataxia and motor learning deficits [37–39], or neuroinflammatory pathways relevant to cerebellar agings [40–42]. Additionally, the protocol enables analyses of synaptic plasticity mechanisms within cerebellar circuits, such as long-term depression (LTD) at parallel fiber-Purkinje cell synapses, which is critical for motor learning and coordination [43–45]. The improved optical properties of OCerSC at 10 DEV (**Fig 2D**), combined with reduced cell death in SF + I conditions (**Fig 2B**, **Supplementary Table 1 in S1 File**), further enhance its utility for advanced imaging techniques, such as confocal microscopy (**Fig 3A**-**3E**) and electrophysiology. For researchers interested in directly visualizing cell death patterns, propidium iodide staining is a well-established complementary approach that can be readily applied to these cultures at desired time points, providing orthogonal validation of the low cytotoxicity observed via LDH release.

Our protocol addresses excessive inflammation through media modification, providing a standardized, reproducible method that adds significant value to the literature. It enables prolonged ex vivo studies of aged cerebellar tissue, opening new avenues for investigating age-related neurodegenerative diseases and developing targeted therapeutic interventions. In conclusion, this innovative approach enhances our understanding of neurological processes and supports future discoveries in neuroscience research.

## Supporting information

**S1 File. Step-By-Step Protocol, Also Available On Protocols.io.**
(PDF)

**S2 File. Supplementary Figure 1 And Supplementary Table 1.**
(PDF)

## Acknowledgments

We thank the Schisler Laboratory and the McAllister Heart Institute administration team members for their fellowship and support. We also thank the UNC Microscopy Services Laboratory, Department of Pathology and Laboratory Medicine, and the UNC Hooker Imaging Core Facility for their support with image acquisition and processing using the Andor Dragonfly and Leica STELLARIS 8 FALCON microscopes. All authors have approved the final version of this manuscript and agree to be accountable for all aspects of the work, ensuring that any questions related to the accuracy or integrity of any part of the work are appropriately investigated and resolved. All individuals designated as authors qualify for authorship, and all those who qualify are listed.

## Author contributions

**Conceptualization:** Kaitlan Smith.

**Formal analysis:** Michael F Almeida, Kaitlan Smith.

**Funding acquisition:** Michael F Almeida, Kaitlan Smith, Jonathan C Schisler.

**Investigation:** Michael F Almeida, Kaitlan Smith, Michael A Garris, Meagan Colie.

**Methodology:** Michael F Almeida, Kaitlan Smith, Meagan Colie.

**Project administration:** Michael F Almeida.

**Supervision:** Jonathan C Schisler.

**Writing – original draft:** Michael F Almeida, Kaitlan Smith, Jonathan C Schisler.

**Writing – review & editing:** Michael F Almeida, Rebekah Sanchez-Hodge, Meagan Colie, Jonathan C Schisler.

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
