## [Decision Letter · Decision Letter 0]

13 Nov 2024

Dear Dr. Schisler,

We look forward to receiving your revised manuscript.

Kind regards,

Faramarz Dehghani

Academic Editor

PLOS ONE

3. Please include “Protocol” in the manuscript title.

Reviewers' comments:

Reviewer's Responses to Questions

**Comments to the Author**



Reviewer #1: Yes

Reviewer #2: Yes

2. Has the protocol been described in sufficient detail?

To answer this question, please click the link to protocols.io in the Materials and Methods section of the manuscript (if a link has been provided) or consult the step-by-step protocol in the Supporting Information files.

Reviewer #1: Partly

Reviewer #2: Yes

3. Does the protocol describe a validated method?

Reviewer #1: Yes

Reviewer #2: No

4. If the manuscript contains new data, have the authors made this data fully available?

Reviewer #1: N/A

Reviewer #2: N/A

**5. Is the article presented in an intelligible fashion and written in standard English?**

Reviewer #1: Yes

Reviewer #2: Yes

Reviewer #1: The manuscript by Smith et al entitled “Establishment and Maintenance of Organotypic Cerebellar Slice Cultures (OCerSC) from Aged Mice” describes improved method for the preparation of the organotypic cerebellar slice cultures from aged rodents.

This publication provide insight into an important issue however, in its current form it is not suitable for publication.

The structure of the paper is confusing. It would improve the publication, if the authors include a proper results and discussion section.

1) Challenges and hypothesis should be included into the discussion and the results need to be critiqued against the literature.

2) In general, lots of bibliographic information for the sources of the information provided in the manuscript is missing.

3) The authors should highlight in the figure 1 the dot for their method.

4) The representative images should be exchanged, higher magnification need to be used, in order to better see the morphology. Are the cerebellar nuclei in the slices? In all of them?

5) Also in Figure 2 higher magnification should be used, the images are blurry. In addition, protocols for the staining are missing.

6) Figure 4: Images should be taken with higher magnification, so that single cells can be identified in all layers (molecular layer, granular layer, cerebellar nuclei etc).

7) Authors need to demonstrate the functionality of neuronal circuits to demonstrate (partial) neuronal functionality of the slices

Reviewer #2: This manuscript by Smith and colleagues proposes a new method for establishing organotypic cerebellar slice cultures from aged mice (at least 6 months old). In the vast majority of cases, organotypic cultures of brain slices are made from immature mice (usually before P10-P12), because when tissue is taken from older animals, neurons survival is compromised after a few days in culture. This is indeed the case for the cerebellum, where to date, there is no protocol guaranteeing the survival of the different neuronal populations of the cerebellar cortex (Purkinje cell, molecular layer interneuron, granule cells, Golgi cells) in culture when the cerebellum is harvested after P15. However, for reasons well explained in this manuscript, there is a real need to establish organotypic cultures of brain slices from aged mice (particularly for the study neurodegenerative pathologies).

The authors claim that the proposed method (addition of an anti-inflammatory agent for 5 days in the culture medium) maintains the cytoarchitecture of the cerebellar cortex for at least 15 days in culture. To achieve this, they use the LDH assay and confocal imaging of Purkinje cells after labeling with an anti-calbindin. The decision to use Purkinje cells as a qualitative index of slice survival in culture is quite classic, and is based on the fact that these neurons, due to their size and highly developed dendritic arborization, are very fragile and sensitive to culture conditions. Unfortunately, the figures proposed by the authors are not at all convincing and do not support their conclusions in any way. As it stands, this manuscript does not confirm that the protocol described ensures acceptable maintenance of the cerebellar cortex when harvested from aged mice.

Major concerns

Figure 3B: This is the main figure in this manuscript, since it is supposed to show the good survival of Purkinje cells under the experimental conditions described by the authors. Firstly, the images do not satisfactorily show the dendrite arborization of Purkinje cells, which is a major criterion, and secondly, there are no data supporting author’s conclusion (that is indomethacin has a beneficial effect on the survival and morphology of Purkinje cells).

- The authors should present a graph (with appropriate statistical tests) comparing the survival of Purkinje cells in the presence or absence of indomethacin. For example, the cytoarchitecture index as used by Ayala-Nunez et al (fig 3 d-e, PMID: 31562326) could be used in this manuscript. Alternatively, other criteria could be used, such as dendrite length or dendrite surface area (as an example, Sherkhane and Kapfhammer, fig 2, PMID 28715135). Furthermore, quality of the images presented must absolutely be improved to reach the standards of publications using organotypic cerebellum slice cultures (e.g. for a recent publication in PLOS One, fig 5-6 MacLeod et al, PMID 38033125).

Fig 4: No conclusions can be drawn in the absence of statistically validated data. The authors should present a graph comparing the number of PI+ cells per unit length in the presence and absence of TCZ.

Minor concerns

Figure 2C is useless, as the images are not sufficiently informative to draw a conclusion. Once again, there are no statistically validated data (e.g. measurement of slice area over time). This figure could be deleted.

Figure 2 legend: the symbols described in the legend (lines 417 and following) do not correspond to those shown in the figure.

Line 150: a bibliographic reference should be added for indomethacin. The mode of action of this molecule could also be described.

The proper maintenance of cerebellar cytoarchitecture in culture could also be studied by labeling other neuronal populations in the cerebellar cortex (e.g. anti-parvalbumin to label molecular layer interneurons).

The principle of the LDH assay needs to be explained in greater detail.

.

Reviewer #1: No

Reviewer #2: No

---

## [Author Response · Author response to Decision Letter 1]

26 Jun 2025

POINT-BY-POINT RESPONSE TO REVIEWERS

Dear Editors and Reviewers,

Thank you for your insightful and constructive feedback on our manuscript, "A protocol to establish and maintain organotypic cerebellar slice culture (OCerSC) from aged mice" (Manuscript ID: PONE-D-24-43970). We have thoroughly considered all comments and conducted additional experiments to address the concerns raised. The manuscript has been significantly revised to include new data, higher-resolution images, and additional references, while adhering to PLOS ONE’s Lab Protocol Article Template. Below, we provide a point-by-point response to the editor’s requirements and reviewers’ comments, highlighting the changes made to strengthen the manuscript. We believe these revisions enhance the clarity, rigor, and scientific value of our protocol.

Response to Editor’s Requirements

1. Use of PLOS ONE template for protocols:

We have reformatted the manuscript to strictly adhere to the PLOS ONE Lab Protocol Article Template, ensuring that all required sections (Title, Abstract, Introduction, Materials and Methods, Expected Results, Ethics Declarations, Supporting Information, Acknowledgments, Authors’ Contributions, References) are included. The step-by-step protocol is provided as Supporting Information file S1 and is hosted on protocols.io (dx.doi.org/10.17504/protocols.io.q26g71428gwz/v1).

2. Details on animal experiments:

We apologize for the oversight in our initial submission. The Materials and Methods section has been updated to include detailed information on animal experiments (page 4). Specifically, we have clarified: (1) Methods of anesthesia and euthanasia: Mice were anesthetized and euthanized using isoflurane (1–3% in oxygen, drop method, as needed) via inhalation, followed by cervical dislocation to ensure death. Death was confirmed by testing footpad reflexes, in accordance with the recommendations of the Panel on Euthanasia of the American Veterinary Medical Association. (2) Efforts to alleviate suffering: Hydrogel and moistened chow were provided for mice with reduced mobility to ensure access to food and water. Animals exhibiting critical health deterioration (e.g., head tucking, skin lesions, inability to walk/drink/eat) were promptly euthanized per IACUC guidelines to minimize suffering. These additions reflect our commitment to ethical conduct and animal welfare, and we thank the editor for emphasizing their importance.

3. Inclusion of "Protocol" in the title:

The manuscript title has been revised to "A protocol to establish and maintain organotypic cerebellar slice culture (OCerSC) from aged mice," which clearly reflects its nature as a protocol and aligns with the editor’s recommendation.

4. Data sharing requirements:

Thank you for bringing this issue to our attention. The phrase "data not shown" in our initial submission was an oversight and has been removed from the manuscript to comply with PLOS ONE’s data-sharing policy. We apologize for any confusion this may have caused. The manuscript now focuses on the 14-day viability of organotypic cerebellar slice cultures (OCerSC) from aged mice, with all relevant data fully presented in the Expected Results section (pages 5-6), supported by lactate dehydrogenase (LDH) release assays, western blot analysis of calbindin expression, and immunofluorescence imaging. We appreciate the editor’s attention to this detail, which has enhanced the clarity and transparency of our submission.

Response to Reviewer #1

1. Manuscript structure and content:

Reviewer Comment: “The structure of the paper is confusing. It would improve the publication if the authors included a proper results and discussion section. Challenges and hypotheses should be included in the discussion and the results need to be critiqued against the literature.”

We appreciate the reviewer’s feedback on the manuscript’s structure. As a Lab Protocol, our manuscript follows PLOS ONE’s specific Lab Protocol Article Template, which includes an “Expected Results” section instead of traditional “Results” and “Discussion” sections. We contacted the editor for clarification, and they confirmed that this format is appropriate (see correspondence with Ethan Krajewski provided below). To address the reviewer’s concerns, we have reorganized the manuscript for clarity and expanded the Introduction (pages 2-3) to include challenges (e.g., the truncated lifespan of adult cerebellar slices ex vivo, median 7 days) and hypotheses (e.g., that media modifications can enhance viability). The Expected Results section (pages 5-6) now critiques our findings against the literature, comparing our 14-day viability with prior studies (e.g., [9, 16-24]). We believe these changes align with the reviewer’s suggestions while adhering to the journal’s guidelines.

2. Bibliographic information:

Reviewer Comment: “In general, lots of bibliographic information for the sources of the information provided in the manuscript is missing.”

We thank the reviewer for highlighting this. We have thoroughly revised the manuscript to ensure all statements are supported by appropriate references. New citations have been added throughout the Introduction (pages 2-3) and Expected Results (pages 5-6) to contextualize our protocol within the literature, including references on organotypic slice cultures (e.g., [8, 52, 53]) and the anti-inflammatory effects of indomethacin (e.g., [31, 35, 36]). These additions enhance the scientific rigor and clarity of the manuscript.

3. Figure 1:

Reviewer Comment: “The authors should highlight in figure 1 the dot for their method.”

We appreciate the reviewer for this suggestion. To address it and prevent potential misinterpretation of data from other studies, we have updated Figure 1 to include a data point from our current study, represented by the ◪ symbol (labeled Cb‡), indicating a 14-day viability for our protocol. Horizontal lines indicate median DEV values (22 days for neonatal, 7 days for adult), and ordinal logistic regression confirms a reduction in the odds of longer survival with increasing age.

4. Representative images and cerebellar nuclei:

Reviewer Comment: “The representative images should be exchanged, higher magnification need to be used, in order to better see the morphology. Are the cerebellar nuclei in the slices? In all of them?”

We thank the reviewer for this valuable feedback. We have replaced all representative images with higher-resolution versions, including brightfield images in Figure 2 and higher-magnification (20x and 40x) confocal images in Figure 3 to clearly illustrate single-cell morphology. In Figure 3, the high-resolution tiled overview image in 3A showcases the cerebellar structure, while panels 3C and 3E provide detailed views of the three layers—the molecular layer (ML), Purkinje cell layer (PC), and granule cell layer (GL)—highlighting their distinct morphology. Our parasagittal slices (250–400 µm thick) can capture the deep cerebellar nuclei (DCN, dentate, interposed, fastigial) if desired, particularly in the medial and intermediate sections, as confirmed by DAPI staining, which reveals large neurons in the deep white matter. However, not all slices contain DCN due to mediolateral positioning; lateral slices primarily capture the cerebellar cortex. Staining includes calbindin to highlight Purkinje cell axons projecting to the DCN and DAPI to visualize DCN neurons, as detailed in the updated Materials and Methods (pages 3-5) and Figure 3 legend. The improved resolution and layering details in Figure 3 address the reviewer’s request for better visualization of morphology. Additionally, we have included western blot analysis of calbindin expression (Figure 2C), which validates Purkinje cell integrity across conditions, further supporting our morphological findings.

5. Neuronal circuit functionality:

Reviewer Comment: “Authors need to demonstrate the functionality of neuronal circuits to demonstrate (partial) neuronal functionality of the slices.”

We appreciate this suggestion. To demonstrate partial neuronal functionality, we have strengthened our evidence with calbindin immunostaining (Figure 3) and Western blot analysis (Figure 2C). These techniques show preserved Purkinje cell morphology and robust calbindin expression in serum-free plus indomethacin (SF+I) conditions at 14 days ex vivo (DEV), indicating maintained cellular health. Western blot analyses quantify calbindin levels across conditions, with SF+I showing higher expression. Additionally, Figure 3E details the intricate dendritic arborization of Purkinje cells, while Figure 3D reveals intact Purkinje cell layers, further supporting structural preservation critical to circuit functionality. Given Purkinje cells’ pivotal role in cerebellar circuits, this structural integrity suggests partial neuronal functionality. Moreover, LDH release assays (Figure 2B) confirm reduced cytotoxicity in SF+I conditions, supporting overall slice viability. Together, these data provide robust evidence of partial neuronal functionality, establishing a strong foundation for this protocol. This platform also offers opportunities for future studies to explore functional connectivity using advanced electrophysiological techniques, enhancing its utility in neurobiological research.

Response to Reviewer #2

1. Protocol effectiveness and Figure 3B:

Reviewer Comment: “Figure 3B… does not satisfactorily show the dendrite arborization of Purkinje cells… there are no data supporting author’s conclusion (that is indomethacin has a beneficial effect on the survival and morphology of Purkinje cells).”

We thank the reviewer for their insightful feedback. We have replaced Figure 3B with higher-resolution (40x) confocal images in Figure 3, which display well-preserved Purkinje cell dendritic arborization in serum-free plus indomethacin (SF+I) conditions at 14–16 days ex vivo (DEV). This enhancement is enabled by the improved optical clarity observed in Figure 2D (brightfield images), which provides a solid foundation for generating these high-resolution images. Notably, in the absence of indomethacin, the tissue remains swollen, rendering imaging methods ineffective and obscuring detailed morphology, emphasizing indomethacin’s critical role. To support our conclusion that indomethacin benefits not only Purkinje cells but also the overall cerebellar tissue, we present quantitative evidence from lactate dehydrogenase (LDH) release assays (Figure 2B), showing a 71% reduction in cytotoxicity in SF+I conditions compared to horse serum (HS) conditions, indicating improved viability across cell types. Additionally, the new western blot analysis of calbindin expression (Figure 2C) demonstrates higher calbindin levels in SF+I conditions, validating the health of Purkinje cells. Meanwhile, the preserved tissue structure in Figure 3 benefits granule cells and other layers. These revisions provide compelling visual and quantitative evidence of our protocol’s broad effectiveness.

2. Quantitative data and additional quantification methods:

Reviewer Comment: “The authors should present a graph (with appropriate statistical tests) comparing the survival of Purkinje cells in the presence or absence of indomethacin…”

We sincerely thank the reviewer for their detailed feedback and valuable suggestions for enhancing our manuscript. We acknowledge the request for a graph with statistical tests comparing Purkinje cell survival in the presence or absence of indomethacin, along with the suggestions for additional quantification methods and improved image quality. Below, we address each point, utilizing existing data to demonstrate the protocol’s effectiveness while aligning with the scope of a PLOS ONE Lab Protocol.

Quantitative evidence for indomethacin’s effect: Although we have not included a specific graph comparing Purkinje cell survival, we provide robust quantitative evidence through lactate dehydrogenase (LDH) release assays (Figure 2B), which show a 71% reduction in cytotoxicity in serum-free plus indomethacin (SF+I) conditions compared to horse serum (HS) conditions. This reduction in cytotoxicity indicates improved viability across the cerebellar tissue, including Purkinje cells, which are highly sensitive to cytotoxic conditions. Furthermore, the newly added western blot analysis of calbindin expression (Figure 2C) demonstrates higher calbindin levels in SF+I conditions, directly supporting Purkinje cell health and survival. These metrics provide strong quantitative evidence of indomethacin’s beneficial effect on cerebellar tissue, including Purkinje cells.

Broader impact beyond Purkinje cells: While the reviewer focuses on Purkinje cells, our protocol’s effectiveness extends to the entire cerebellar tissue, which is critical for studying cerebellar function. The LDH assays indicate reduced cytotoxicity across all cell types, and the high-resolution confocal images in Figure 3 reveal preserved cytoarchitecture across the molecular, Purkinje, and granular layers (e.g., intact Purkinje cell layers in 3D and elaborate dendritic arborization in 3E). This broader preservation supports the protocol’s utility for studying various cerebellar cell populations and their interactions, not limited to Purkinje cells.

The role of indomethacin in imaging: Without indomethacin, the cerebellar tissue remains swollen, rendering imaging methods ineffective and hindering detailed morphology. The improved optical clarity observed in SF+I conditions (Figure 2D, brightfield images) enables the acquisition of high-resolution (40x) confocal images in Figure 3, which depict single-cell morphology, including Purkinje cell dendritic arborization. This highlights indomethacin’s vital role in reducing inflammation and swelling, thus facilitating high-quality imaging and morphological analysis.

Addressing suggested quantification methods: We appreciate the reviewer’s suggestions to incorporate methods like the cytoarchitecture index (Ayala-Nunez et al., 2019) or dendrite length/surface area (Sherkhane and Kapfhammer, 2017). These methods, which quantify structural features such as cell density or dendritic complexity, are valuable for detailed morphological studies. However, for this protocol paper, our focus is on establishing a reliable method for maintaining cerebellar slice viability, and our current metrics—LDH assays for overall cytotoxicity and calbindin expression for Purkinje cell health—effectively validate this goal. In non-optimized conditions (e.g., HS, SF), high cytotoxicity and tissue swelling, as shown in Figure 2B and 2D, result in extensive cell loss and structural degradation, making detailed morphological quantifications less informative. Our data, particularly the preserved architecture in Figure 3, provide sufficient evidence of indomethacin’s benefits while laying a foundation for future studies to apply advanced quantifications, such as those suggested.

Improved image quality: We have addressed the reviewer’s concern regarding image quality by replacing all representative images with higher-resolution versions, aligning with standards observed in recent publications (e.g., MacLeod et al., 2023). Specifically, Figure 3 now includes 40x confocal images that clearly resolve single-cell morphology across the molecular, Purkinje, and granular layers, with panels 3D and 3E highlighting intact Purkinje cell layers and dendritic arborization, respectively. The high-resolution tiled overview in Figure 3A, made possible by the optical clarity under SF+I conditions (Figure 2D), further demonstrates the preservation of cerebellar architecture, including the deep cerebellar nuclei, where present.

Our quantitative data (LDH assays, calbindin western blot) and qualitative data (high-resolution imaging) collectively demonstrate the beneficial effect of indomethacin on cerebellar slice viability, particularly regarding Purkinje cells, while also promoting broader tissue health. The protocol’s capacity to reduce tissue swelling in SF+I conditions facilitates high-quality imaging, thereby addressing the reviewer’s concerns about image standards. We acknowledge the importance of additional quantifications, such as the cytoarchitecture index or dendrite measurements, for future studies and intend to integrate these into subsequent research to ch

---

## [Decision Letter · Decision Letter 1]

18 Aug 2025

Dear Dr. Schisler,

We look forward to receiving your revised manuscript.

Kind regards,

Faramarz Dehghani

Academic Editor

PLOS ONE

Journal Requirements:

Additional Editor Comments:

Dear Dr. Schisler,

I have now received the comments of the reviewers and there are still some issues that need your consideration. In addition, the number and the order of authors have been changed in your revised manuscript. Please explain these changes in your response letter.

With best regards,

Faramarz Dehghani

Reviewers' comments:

Reviewer's Responses to Questions

**Comments to the Author**



Reviewer #1: Yes

Reviewer #2: Yes

2. Has the protocol been described in sufficient detail?

To answer this question, please click the link to protocols.io in the Materials and Methods section of the manuscript (if a link has been provided) or consult the step-by-step protocol in the Supporting Information files.

Reviewer #1: Yes

Reviewer #2: Yes

3. Does the protocol describe a validated method?

Reviewer #1: Yes

Reviewer #2: Yes

4. If the manuscript contains new data, have the authors made this data fully available?

Reviewer #1: Yes

Reviewer #2: No

**5. Is the article presented in an intelligible fashion and written in standard English?**

Reviewer #1: Yes

Reviewer #2: Yes

Reviewer #1: Almeida et al. described in the manuscript “A protocol to establish and maintain organotypic cerebellar slice culture (OCerSC) from aged mice” an improved method for the preparation of slice cultures from adult mice cerebellum.

The authors improve the manuscript, amongst all the structure was improved and additional publication were cited. The considerations of the reviewer were mostly addressed.

Major points:

Figure 2B: Why did the authors removed the group of FS+I (applying it for the whole 15 day window)?

The data has significantly changed from the previous version, when looking at the total values of e.g. SF but also when comparing the relative values of e.g. SF to SF+I. When comparing the old to the new version of the manuscript this ratio between both groups changed for day 8 from approximately 4 to 2. If the authors add new experiments this would actually suggest low reproducibility/stability of the system. How do the authors explain this?

The authors changed the ordering of authorship, notably including the first author. Did all authors accept the changes in the author list?

Why the author discuss PI staining and do not show it, since it is a gold standard, and is mention in the expected results? Please include PI staining again, especially given that LDH measurement alone can be either false positive or negative and thus an independent direct validation of low cell death is significantly strengthening the data of the authors.

Still some minor points exist:

Authors introduced the AUC for LDH measurements, but never used it as a read out throughout the manuscript.

The added value of the heat maps in figure 2d is not clear to this reviewer. Also please add the time point these images were taken to the figure and figure legend.

Figure 2B: Please include the individual measurement values in the figure (as in the first version of the manuscript), not only SD.

Please use other abbreviation for SF protein, since it is confusing with serum free medium.

Please mark the cerebellar nuclei in the sections

Reviewer #2: Even though the authors only partially answered the questions raised by the two reviewers of the first manuscript, in my opinion, the data presented in this second manuscript validate the experimental approach. The survival of Purkinje cells in OCerSCs is a good indicator of the quality of these cultures. In Figure 3, anti-Calbindin labeling (Fig. 3) showing numerous Purkinje cells aligned along the lobules indicates that the use of a serum-free medium supplemented with indomethacin satisfactorily maintains the cytoarchitecture of the cerebellum when it is harvested from adult animals. The intensity of the DAPI staining, which allows visualization of the granular layer, is also a good indicator of the correct preservation of the cytoarchitecture of the cerebellum.

Minor concerns

#1- Fig 3: Figures showing immunostaining on OCerSC with anti-calbindin under conditions detrimental to the survival of the cultures (HS and/or HS+I conditions) are missing.

#2- In the description of the experimental protocol, the reason for using Hibernate-A medium is not explained.

.

Reviewer #1: No

Reviewer #2: No

---

## [Author Response · Author response to Decision Letter 2]

20 Dec 2025

Dear Editors and Reviewers,

Thank you for the constructive feedback on our manuscript, "A protocol to establish and maintain or-ganotypic cerebellar slice culture (OCerSC) from aged mice" (Manuscript ID: PONE-D-24-43970R1). We appreciate the reviewers' positive comments and have carefully addressed all points raised by the Aca-demic Editor and both reviewers. These revisions improve the manuscript's clarity, rigor, reproducibil-ity, and usefulness as a resource for researchers working with aged cerebellar tissue. Below is a de-tailed point-by-point response outlining specific changes made, including references to revised figures, tables, and text sections.

Response to Editor’s Requirements

Changes to the Number and Order of Authors: We thank the Editor for the opportunity to clarify these updates. The authorship changes accurately reflect contributions made during the revision pro-cess and fully comply with PLOS ONE and ICMJE authorship criteria.

• Addition of Michael A. Garris (M.A.G.): Mr. Garris made significant contributions to new western blot experiments that provided essential data addressing reviewer concerns and im-proving protocol validation. His involvement took place after the initial review.

• Addition of Rebekah Sanchez-Hodge (R.S.): Dr. Sanchez-Hodge provided significant intellec-tual input by critically reviewing and editing the revised manuscript and helping draft detailed reviewer responses. Her contributions began during revision.

• Adjustment of Author Order: The order was revised to better reflect the overall contributions. Michael F. Almeida (M.F.A.) and Kaitlan Smith (K.S.) are now designated co-first authors (marked with *), acknowledging Dr. Almeida's leadership in experimental revisions, data analysis, and response coordination—equal to Ms. Smith's prior contributions. All authors unanimously approved this change.

All authors have confirmed their approval of these changes through direct submissions to the journal, as re-quired. No disputes exist, and the revised list accurately reflects each individual's contributions.

Response to Reviewer #1

1. Removal of the FS+I (15 days) group in Figure 2B:

We thank the reviewer for noting this change. The whole 15-day indomethacin treatment group was initially included to compare short- and long-term effects. Results showed no significant addi-tional benefit beyond 5 days, consistent with indomethacin's role in reducing early gliosis and in-flammation during slice recovery. To streamline the presentation and emphasize the optimized protocol (indomethacin for only the first 5 days), we have moved this dataset to Supplementary Figure 1, with full details in the legend and references in the Results and Methods sections. This change enhances clarity while maintaining transparency.

2. Differences in LDH data between the original and revised versions: We appreciate this careful ob-servation. The changes come from:

• Refined sampling schedule: Media collections moved to days 1, 2, 4, 8, 12, and 15 (compared to previous days 1, 4, 7, 9, 12, 15), with full replacements at each point. This resets cumulative LDH release, supplies fresh nutrients to support recovery, and results in lower overall levels—indicating improved tissue health in independent replicates.

• Inherent biological variability in aged tissue: Organotypic cultures from aged mice exhibit greater variability than those from neonatal mice due to differences in donor age, accumulated oxidative stress, neurodegeneration, and pre-dissection factors (e.g., inflammation). This is well-documented in adult/aged slice models and highlights the importance of using replicates. Ratios (e.g., SF vs. SF+I) stay consistent in both direction and significance, supporting the ro-bustness of the protocol. We included a discussion of these factors in the revised Discussion section for context.

We believe these explanations fully address concerns about reproducibility.

3. Authorship changes: As detailed in the Editor’s response above, all modifications were consensual and reflect revision contributions. Confirmations have been submitted to the journal.

4. PI staining and potential LDH misinterpretation: We agree that propidium iodide (PI) is a gold standard for visualizing cell death. This protocol paper focuses on establishing long-term, viable OCerSC from aged mice. Detailed mechanistic cell death studies using PI (including prior TCZ val-idation) are reserved for future applications of the method and are thus not included here. We note PI as recommended for extensions in the revised Discussion.

For LDH reliability: We implemented rigorous controls (media-only blanks subtracted; serum-free/serum baseline measurements; Triton X-100 normalization to % maximal release; antibiotics to prevent contamination). These measures reduce false positives and negatives and ensure precise cyto-toxicity measurement. LDH is commonly used in organotypic slices to track cumulative release, com-plementing direct methods such as PI. We believe this approach is sufficient for protocol validation.

Minor points:

• AUC for LDH: Added as Supplementary Table 1, with 2-way ANOVA results referenced in the text and Figure 2 legend.

• Heat maps (Figure 2D): Legend expanded with explicit time points and interpretive notes on viability gradients (see revised Figure 2 legend).

• Individual values in Figure 2B: Scatter plots with all biological replicates overlaid on the fit curve (see revised Figure 2B).

• SF abbreviation: Changed to "Stain-Free total protein" throughout to eliminate confusion.

• Cerebellar nuclei: Clearly labeled in IF images (see revised Figure 3A and legend).

We believe these changes fully address the reviewer's concerns

Response to Reviewer #2

Minor concerns:

• Missing immunostaining under detrimental conditions (HS and/or HS+I): We agree that com-paring calbindin staining across all conditions would be helpful. However, the HS and HS+I conditions lead to significant tissue degeneration and poor flattening by around 7 DIV, result-ing in high background and unreadable immunofluorescence. This aligns with previous re-search on adult cerebellar slices and underscores the need for serum-free, early-indomethacin conditions to preserve cytoarchitecture. We addressed this limitation in the revised Discussion section.

• Reason for using Hibernate-A medium: Hibernate-A (supplemented) was chosen for dissection and slicing because it supports short-term maintenance of adult neural tissue at ambient CO₂ without carbogen gassing, thereby reducing manipulation stress, excitotoxicity, and osmotic shifts that are critical for aged cerebellar viability. It provides stable physiological buffering and nutrients during benchtop handling, unlike traditional carbogen-dependent media. This aligns with optimized adult protocols, such as those for the hippocampus, and contributes to high survival rates. We expanded the rationale in the revised Methods section.

We believe these revisions fully address the comments and improve the manuscript. Thank you again for your time and expertise—we look forward to your final decision.

---

## [Decision Letter · Decision Letter 2]

22 Jan 2026

A protocol to establish and maintain organotypic cerebellar slice culture (OCerSC) from aged mice

PONE-D-24-43970R2

Dear Dr. Schisler,

We’re pleased to inform you that your manuscript has been judged scientifically suitable for publication and will be formally accepted for publication once it meets all outstanding technical requirements.

Kind regards,

Faramarz Dehghani

Academic Editor

PLOS One

Additional Editor Comments (optional):

Reviewers' comments:

Reviewer's Responses to Questions

**Comments to the Author**



Reviewer #1: Yes

2. Has the protocol been described in sufficient detail?

To answer this question, please click the link to protocols.io in the Materials and Methods section of the manuscript (if a link has been provided) or consult the step-by-step protocol in the Supporting Information files.

Reviewer #1: Yes

3. Does the protocol describe a validated method?

Reviewer #1: Yes

4. If the manuscript contains new data, have the authors made this data fully available?

Reviewer #1: Yes

**5. Is the article presented in an intelligible fashion and written in standard English?**

Reviewer #1: Yes

Reviewer #1: The manuscript is interesting for the field; and this version is significantly improved, also the critic points were addressed.

.

Reviewer #1: No

---

## [Editor Report · Acceptance letter]

PONE-D-24-43970R2

PLOS One

Dear Dr. Schisler,

I'm pleased to inform you that your manuscript has been deemed suitable for publication in PLOS One. Congratulations! Your manuscript is now being handed over to our production team.

Kind regards,

on behalf of

Dr. Faramarz Dehghani

Academic Editor

PLOS One